# Effect of Scarf Repair Geometry on the Impact Performance of Aerospace Composites

**DOI:** 10.3390/polym15102390

**Published:** 2023-05-20

**Authors:** Sridharan Vijay Shankar, Sridhar Idapalapati

**Affiliations:** 1Rolls-Royce @ NTU Corporate Laboratory, Nanyang Technological University, Singapore 639798, Singapore; sridhara001@e.ntu.edu.sg; 2School of Mechanical and Aerospace Engineering, Nanyang Technological University, Singapore 639798, Singapore

**Keywords:** scarf patch, film adhesive, secondary bonding, impact loading, failure analysis

## Abstract

This experimental study investigates the effect of scarf geometry in restoring the impact response of scarf-patched 3 mm thick glass-fiber reinforced polymer (GFRP) matrix composite laminates. Traditional circular along with rounded rectangular scarf patch configurations are considered repair patches. Experimental measurements revealed that the temporal variations of force and energy response of the pristine specimen are close to that of circular repaired specimens. The predominant failure modes were witnessed only in the repair patch which includes matrix cracking, fiber fracture, and delamination, and no discontinuity in the adhesive interface was witnessed. When compared with the pristine samples, the top ply damage size of the circular repaired specimens are larger by 9.91%, while that of the rounded rectangular repaired specimens is larger by 434.23%. The results show that circular scarf repair is a more suitable choice of repair approach under the condition of a 37 J low-velocity impact event even though the global force-time response is similar.

## 1. Introduction

The continual drive for re-engineering of glass fiber reinforced polymer (GFRP) based load bearing components has opened avenues for design for repair. Especially, when load-bearing composite components operate in diverse conditions, they are susceptible to foreign object impact during the operational stage as well as accidental tool drop impacts during maintenance, repair, and overhaul (MRO) events [1]. These accidents can induce barely visible damages to laminate delamination initiated by matrix cracking and its channeling through fibers or concentrated commuted fracture, consequently reducing laminate strength and stiffness [2]. Particularly, low-velocity impact damage includes different damage mechanisms, such as delamination, matrix cracking, fiber failure, and through penetration. Panettieri et al. [3] analyzed low energy impacts from 4.5 J to 30 J on quasi-isotropic carbon fiber reinforced polymer (CFRP) with different thicknesses. They pointed out that large contact force fluctuations were due to delamination onset, but the subsequent ultrasonic scans did not reveal any significant internal damage. Mehmet et al. [4] investigated the impact response of unidirectional GFRP and further discussed damage modes and damage processes based on a variation of absorbed energy (*E_a_*) and impact energy (*E_i_*). They found that the primary damage mode for higher impact energies was found to be fiber fracture whereas, for smaller impact energies it was indentation resulting in delamination and matrix cracks. Koloor et al. [5] found that delamination of laminate lowers flexural strength up to 46.7% compared to their pristine sample, when they are under transverse loading. Hall et al. [6] found that the delaminations formed between plies of different fiber orientations tend to grow along the orientation of the ply beneath the delamination.

Selective parameters that influence the restoration of structural properties includes impact energy, impact location within the repaired area, type of bond, and shape of the repair patch. The effect of impact energy and impact location on the size of the composite’s damage area was extensively studied in the past [7]. The patch shape can influence the reduction in the stress intensity factor and hence restoration of the structural strength [8]. Mechanical fastening and adhesive bonding are the two main approaches used to repair damaged structures to restore them to ultimate design strength. For aircraft structures, an adhesively bonded scarf joint is typically used, as it does not produce protrusions that disrupt the aerodynamic performance of the structure [9] and minimizes joint eccentricities. For identical adherents, scarf joints use relatively small scarf angles to transfer loads efficiently as it distributes shear stresses uniformly in the adhesive layer and low peel stresses at the joint interface and making them a viable MRO option for repairing composite structures [10]. Pengcheng et al. [11] in their work demonstrated maximum tensile strength could be achieved if the variation between the fiber orientations of the patch and the laminate is zero. Similarly, Gursahib et al. [12] in their work determined that the repair of main load-bearing plies (0°) of parent laminate with [45]_2_ patches presented the most favorable residual tensile behavior by effectively releasing the stress concentration in the damaged area.

Significant changes in tension, compression, and impact properties have been observed across different scarf angles [13]. The analysis showed that the increase in scarf angle led to a decrease in tensile and compressive strength. However, reducing the scarf angle excessively is undesirable, as it can be detrimental to the restoration of structural properties by removing excessive material from the parent laminate [13]. In order to find the optimal design space for the scarf angle, an equilibrium between laminate limiting strength to adhesive limiting strength must be determined [14].

Considering the relevance of the repair of composite materials with scarfing in the current aerospace and marine industry, it is necessary to adopt appropriate scarf patch shapes to restore the damaged structure. Popular repair profiles like circular repair patches [8,15,16] have shown promise in reinstating damaged structures. However, they come with the drawback of removing a greater volume of undamaged regions, which adds to the time required for the repair. Previous studies have explored various repair shapes, such as elliptical [8,16], square [8,15], rhombus [15], oval [16], and hexagon [15,16] were conducted. However, no experimental study has been performed to understand the influence of a large sample size (250 mm side square) on low-velocity impact and level of strength restoration when different scarf geometries with the same patch repair volume. Therefore, in this experimental research, we investigate the impact response of 3 mm thick GFRP laminates repaired by circular and rounded rectangular scarf patches with equal repair/patch volume in a double picture frame fixture with a large unsupported-to-support span ratio. It is to be noted that there was no initial damage induced onto the composite laminates for the repair, rather the central region of pristine samples was machined off as per the required dimensions (as detailed in Section 2) and repaired with appropriate scarf patch repair. These were subjected to 32 J impact energy in a drop-weight machine.

## 2. Materials, Manufacturing, and Repair Design

This section details the materials involved in the manufacturing of the laminates and determination of design space for calculation of scarf angle.

### 2.1. Fabrication of FRP Laminates

HexPly M21/37%/7581 GFRP prepregs supplied by Hexcel Ltd.^®^ were used to manufacture the composite laminates. The E-glass prepreg is an 8-harness satin (8HS) weave has 476 g/m^2^ areal density with 50% fiber volume fraction and the cured ply has 0.25 mm thickness. The prepregs layered in the sequence [+45/0/90/−45/0/+45]_s_ were vacuum debulked to remove the trapped air and to achieve a nearly void-free laminate of 3 mm nominal thickness using 12 plies. Herein, the 0° is considered to be the prepreg warp direction of the roll, and 50% fiber volume along warp and weft directions is expected to give the in-plane quasi-isotropy. Finally, it was cured in an autoclave under 0.1 MPa vacuum (to keep the debulking for voids removal), a consolidation pressure of 0.7 MPa, and held at 180 °C temperature for 120 min. Similarly, the repair patch was manufactured with the same fiber orientation and autoclave curing program as that of the parent laminate.

### 2.2. Design and Repair of Laminates

Determination of the optimal scarf angle was adopted from our previous work as summarized by Prabhu et al. [17], where the authors have employed the same HexPly M21/37%/7581 GFRP prepreg and strong, high-temperature resistant AF3109-2K thermosetting modified epoxy scrim supported structural film adhesive (supplied by 3 M) with 60 MPa tensile strength and 2.8 GPa Young’s modulus (mode I and mode II fracture toughness measured under double cantilever beam (DCB) and end notch flexural (ENF) tests was 1.71 kJ/m^2^ and 6.92 kJ/m^2^, respectively). The optimal scarf angle range for the three-dimensional repair of the 3 mm thick laminate was determined to be between 2.66–4.82°. The required scarf angle to meet the design’s ultimate strength requirement is typically small for highly loaded composite structures. However, using a small scarf angle results in very high-stress concentration at the feathered tip and low residual strength unable to meet the design limit load (DLL). In addition to that, shallow or small scarf angles also require greater material removal from the undamaged region which can substantially increase machine time and the possibility of machine-induced damage.

Therefore, from the limits determined, a scarf angle of 3° was chosen. The manufactured laminates were machined using an abrasive water jet to dimensions of 250 × 250 × 3 mm. To repair the damaged region, it must be completely removed and encapsulated within the cut-out of a lower patch, which dictates the shape of the upper repair region. Wang and Gunnion [14] developed a numerical study to present an optimum scarf shape. Their results revealed that significant savings can be made to the amount of material removal (between 26% and 76%), by adopting optimum repairs over conventionally designed repairs under in-plane loading conditions. Furthermore, for low scarf angles, their optimum scarf shape resulted in close to a concentric ellipse with the aspect ratio being approximately equal to the biaxial stress ratio. Additional results also indicated that a hybrid square ellipse profile for high aspect ratio damage could further reduce repair size. Figure 1a illustrates typically seen elongated damaged profiles and commensurate circular and rounded rectangular designs. A rounded rectangular repair patch has 27.5% less volume than a circular repair patch, which also translates to lower undamaged material removal during the machining process.

Three samples tested under pristine (A, B, C), circular patch (D, E, F), and rounded rectangular patch (G, H, I) joined categories. In this study, our objective is to understand the impact performance of circular and rounded rectangular patches bonded with the same volume of material removed during the joining process while machining a 3° scarf angle on a 250 × 250 × 3 mm (Figure 1b,c). As such there was no initial impact damage created on the neat samples, rather to simulate the circular or rounded rectangular patch bonded joints in scarf configuration, pristine samples are machined as required. The approach for scarf repair involves machining the laminates with the help of a pneumatically driven step sander with jigs which facilitates the localized repair. A 7 mm 85/100 Diamond Surface Planer was chosen to sand the surface of the composite. Its high roughness index ensured there is proper mechanical anchorage, as well as there, is no machine-induced delamination. The adherend bonding surface was dry grit blasted for 30 ± 5 s at 250 kPa with 120 μm white alumina powder, which gave a surface roughness (R*_a_*) of 3.40 μm and water contact angle of 45°. After cleaning with deionized water, the samples were dried inside an air-circulating oven for 30 min at 60 °C and degreased with acetone before layering with a single film of AF3109-2K. The overall bonding area for the circular scarf and rounded rectangular scarf repaired samples was computed to be 19,315.13 mm^2^ and 19,317.9 mm^2^, respectively. To avoid film adhesive crimping during the application, the bonding zone was divided into four sectors, and the adhesive film was cut to the shape of the four sectors. After the application of adhesive film in the cavity the patch was assembled on it, ensuring that the fiber orientation of the patch matches with the parent laminate. The repair-patch-cavity assembly was then debulked for a duration of 20 min before the repair After the debulking process, the single layer adhesive film in between the adherend surfaces was cured using the Heatcon^®^ HCS9200B dual zone hot bonder.

## 3. Experimental Methods

A cone-shaped steel impactor with a diameter of 25 mm hemispherical head shown in Figure 2a was used as the drop weight impactor. An accelerometer was fastened to the 3.5 kg impactor to measure the acceleration of the impactor during the testing. The experimental setup of the tool drop test is shown in Figure 2b, the outer 12.5 mm width of the specimen is clamped by a hollow double picture frame such that the central 225 × 225 mm area is not supported. The impactor was allowed to slide freely inside a PVC guide tube at a predetermined height. A slit was cut on the guide tube for the wires on the accelerometer to move freely with the impactor. All specimen edges were fixed with a picture frame on a metal surface plate. The larger side of the repair patch was facing the impactor. A force sensor was sandwiched between a surface plate and a heavy metal base to obtain the impact force. An oscilloscope was used to collect data from the force sensor and the accelerometer. A high-speed camera was placed at the same height as the panel surface to record the impact process and calculate impact velocity and rebound velocity. The initial impact velocity was estimated based on the potential energy of the impactor. The actual initial impact velocity (*v_i_*) and rebound velocity (*v_r_*) are obtained using high-speed camera images. Therefore, the absorbed energy *E_a_* is given as:(1)$$E_{a}=\frac{1}{2}m({v_{i}}^{2}-{v_{r}}^{2})$$
where *m* is mass of the impactor. The velocity during the impact is calculated using:(2)$$v\left(t\right)=v_{i}-\int_{0}^{t}a(t^{\prime })dt^{\prime }$$
where *a*(*t′*) is the acceleration data from the accelerometer attached to the impactor. The displacement-time history *δ*(*t*) is calculated by Newton’s second law as:(3)$$\delta \left(t\right)=v_{i}t-\int_{0}^{t}\left[\int_{0}^{t^{\prime }}a(t^{\prime })dt^{\prime }\right]dt$$

In the current impact experiments, the impactor drop height was set to 1.07 m, resulting in an impact velocity of 4.61 m/s and loading the samples at 37 J. This is a typical high energy (beyond 25 J), these tests were conducted as part of evaluating component repair during the event of accidental tool drop during MRO procedures [4].

## 4. Results and Discussion

The force-time history of the impacted samples is plotted in Figure 3, as the impactor impinges the test sample initially it can be observed that the load increases sharply due to Hertzian contact and has a sudden drop. The Hertzian contact force of pristine and circular repaired samples is (3.68 kN) and that of the rounded rectangular repair sample is 3.87 kN which is 5.1% higher than the pristine sample. Hirai et al. [18] in their work which dealt with the impact response of the woven fabric found that the incipient damage load in their tests on woven GFRP laminates was a consequence of interface failure or matrix cracking near the tension dominant side (bottom side) of the laminates and is given in terms of mode-II fracture toughness (*G_IIc_*) as [19] (the predictions of threshold load value for damage initiation from this simplified model were shown to be in good agreement with experimental load values that correspond to the initial occurrence of damage [20]),
(4)$$P^{2}{}_{cr}=\frac{8\pi^{2}Et^{3}G_{IIc}}{9(1-\nu^{2})}$$
where, *E* (24 GPa) and *υ* are the flexural moduli, Poisson’s ratio of the laminate, *t* is the thickness, and *G_IIC_* is the mode-2 fracture toughness of the laminate (4.55 kJ/mm^2^ [21]). Through Equation (4), it was found that the critical failure load (*P_cr_*) for the GFRP laminates used was 5.1 kN. The peak force observed from the tests is listed in Table 1 (an average of three samples): pristine samples are at 8.39 kN, circular patch repair samples are at 7.72 kN, and rounded rectangular ones are at 7.14 kN. The oscillations within the Hertzian contact and subsequent rise of load can be attributed to impactor ringing and flexural vibration of the impacted specimen. After the Hertzian contact, the load exceeded the *P_cr_* along with large oscillations to the peak observable load. Following peak load, the impactor was allowed to rebound three times till the energy completely dissipated and was able to mimic accidental tool drop in the targeted component.

Typical impact energy versus time responses is depicted in Figure 4. Within experimental scatter, the peak energy absorbed by all three samples is around 33.5 J, close to that of 37 J potential energy imparted by the drop-weight. It is to be clarified that the energy plot in Figure 4 has been plotted for the total duration of the experiment. In contrast, in Figure 3 the impact event starts at ~1.5 ms and ends at about 9.5 ms with the impact event duration being approximately 8.5 ms. According to the energy profile diagram (EPD) proposed by Liu [22], the closeness of impact and the absorbed energy indicate the laminate failure by projectile through penetration. It can be observed that the rounded rectangular repaired sample (I) has absorbed 52% more energy compared to pristine and 36% more compared to circular scarf repaired laminates post-peak energy at the point of impact. High energy absorption by the rounded rectangular scarf repair can directly correlate to the damaged area (Figure 5), which will be discussed in Section 4.1. Similarly, the energy absorbed by the circular scarf repaired samples post-peak energy at the point of impact is greater than pristine while its damage area (Figure 5) is also larger than pristine.

### 4.1. Failure Analysis

Following impact tests, the samples were retrieved for failure examination and was observed that all the samples impacted at 37 J had a damaged region which included a centrally depressed region/permanent indentation, matrix cracking, localized delamination areas, and fiber fracture. Figure 5 shows visually observed impact damage on the three tested specimens. Prior to taking images (with a Nikon Z camera with NIKKOR Z 35 mm f/1.8 S lens on a manual focus setting) the samples were coated with black ink and wiped such that the intricate cracks were filled with isopropanol-based graphite ink and the samples were dried in a convection oven at 85 °C. The samples were placed in the picture frame with high-intensity white light behind the sample. The translucency of the samples enabled us to discern discontinuous fibers as well as matrix and black ink helped accentuate the difference more lucidly. The damaged region was cropped with greyscale inversion to discern the surficial cracks from the undamaged surface.

The damage region of the pristine and circular scarf repair specimens are relatively symmetrical quatrefoil and trefoil shapes, respectively, whereas rounded rectangular scarf repair specimens are asymmetrical with predominant delamination towards the major axis of the rounded rectangle. The damage area was calculated in such a way that the point of impact was considered as the center of damage and the crack that propagated to the maximum length from the center is the radius of the circle of damage. The area of the circle is what gives us the damaged area. The reason for the circle being considered as the damaged area is that one must machine this area for repair purposes. Thus, the damaged area is computed as a circular region whose radius is the furthest point of the delamination region from the impact point. The damage size in circular and rounded rectangular repairs is on average greater by 9.91% and 434.23% when compared to the pristine sample.

The damage areas are approximately 181.63 mm^2^, 199.64 mm^2^, and 970.33 mm^2^, for pristine, circular, and rounded repair specimens, respectively. It is also noticed that the size of the damage is influenced by the patch symmetry, as depicted in Figure 1, despite circular and rounded rectangular scarf repaired samples having similar bonding areas of 19,315.13 mm^2^ and 19,317.9 mm^2^, respectively. The damages witnessed in rounded rectangular scarf repaired laminates are oriented towards the major axis of the patch. In contrast, the damage propagation in minor axes is limited within the bottom cut-out.

Tested samples were further inspected with the help of Nikon XTH 225 ST computed tomography (CT), where the raw images were reconstructed by segmenting the volume. From the scanned images, voxels (volumetric elements) were adjusted by changing the isosurface setting to visualize the impact damages and fiber yarn. Figure 6 illustrates the damaged area relative to the size of the top ply across various plies. Figure 7 illustrates the damage propagation across various plies in through thickness. It can be seen that the pristine and circular scarf patch repaired laminate (Figure 7a,b) demonstrated a similar pine tree pattern [23], whereas in rounded rectangular repaired laminate (Figure 7c) had a cylindrical pattern. These results can also be translated as degradation of bending stiffness with change in the sample configuration [24] i.e., a pristine sample has the highest bending stiffness amongst tested samples followed by circular and rounded rectangular repaired scarf patch laminates.

Figure 8 and Figure 9 show CT scan image slices in the top, third, sixth, ninth, and bottom plies, and depicts the cross-sectional view of the damaged laminates. We observe that the damage size from top to bottom ply gradually increases for both pristine and circular patch-repaired laminates. In contrast, it is almost constant for the rounded rectangular patch laminates. In both pristine and circular repaired patch laminates the evidence of fiber breakage can be witnessed in the plies beneath ply 6 (Figure 9a) and ply 5 (Figure 9b), respectively, in the case of rounded rectangular patch repaired laminate the onset of fiber failure is witnessed from ply 3 onwards (Figure 9c). Figure 8 reveal that all three tested sample undergo permanent indentation at the point of impact. The permanent indentation in the rounded rectangular scarf repaired laminate is most obvious due to maximum roving failure at the bottom side of the laminate enabling greater projectile penetration. It is also noted that the strong aerospace grade structural adhesive AF3109-2k used in this work, was able to eliminate cohesive failure of the adhesive as its mode II fracture toughness of 6.92 kJ/mm^2^ is 1.5 times higher than that of laminate material, and the surface preparation able to resist adhesive failure, under 32 J impact energy.

## 5. Conclusions

This work was carried out to further explore and experimentally validate alternative scarf cut-out shape options against conventional circular scarf cut-outs for localized repairs that have greater material removal and proportionate machining time penalty. An optimal scarf cut-out shape that sufficiently encompasses the damaged region would reduce the downtime of the damaged component along with optimal usage of repair materials, thereby making this a sustainable engineering approach. A critical assessment of current scarf repair design methodologies has revealed that significant savings can be made to the amount of material removal when adopting rounded rectangular repair over conventionally designed circular scarf repaired laminate. Through impact testing, it was revealed that the repairs conducted with similar material volume removed demonstrated different severity of failure modes governed by the symmetricity along major and minor axes. The impact resistance of rounded rectangular scarf repair laminate was only 12.9% lower compared to the pristine whereas circular scarf repair laminate was only 3% lower compared to the pristine. From the CT scan and visual inspection, it was evident that the rounded rectangular scarf repaired laminate incurred maximum damage compared to the circular repair scarf and pristine laminates. However, all three absorbed similar energy during the impact event. It was observed that the energy absorbed by the samples during the impact event was close to the impact energy of the projectile, which was clear as all the samples demonstrated fiber failure and permanent indentation. The rounded rectangular repaired laminates demonstrated maximum permanent indentation, whereas the circular scarf-repaired, and pristine laminates had the least permanent indentation. This research concludes that a localized repair with a rounded rectangular scarf repair has small depreciation in peak impact loading but with a greater damage area can still be a viable repair option considering the downtime and material savings. The strong and tough structural film adhesive AF3109-2K was able to resist the adhesive failure for both cases.

## Figures and Tables

**Figure 1 polymers-15-02390-f001:**
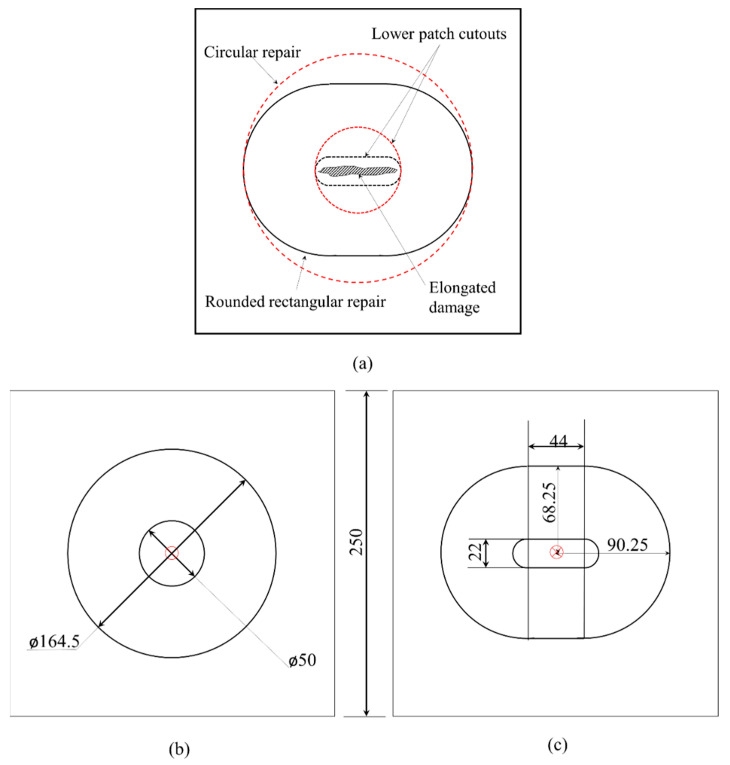
(**a**) Aerial difference between circular and rounded rectangular repairs for same elongated damage profile. Patch designs for a 3° scarfed repair with similar volume of material removed in (**b**) circular, and (**c**) rounded rectangular profiles (all dimensions are in mm).

**Figure 2 polymers-15-02390-f002:**
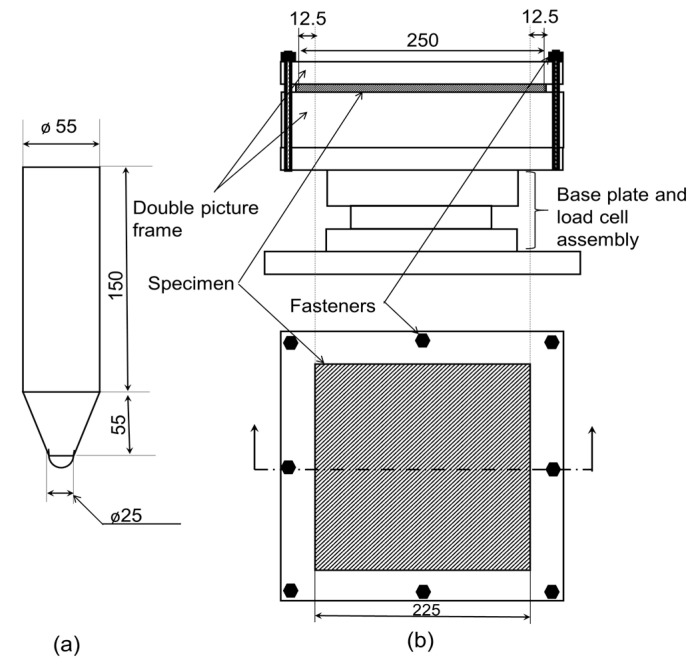
(**a**) projectile profile and (**b**) impact fixture setup (all dimensions are in mm).

**Figure 3 polymers-15-02390-f003:**
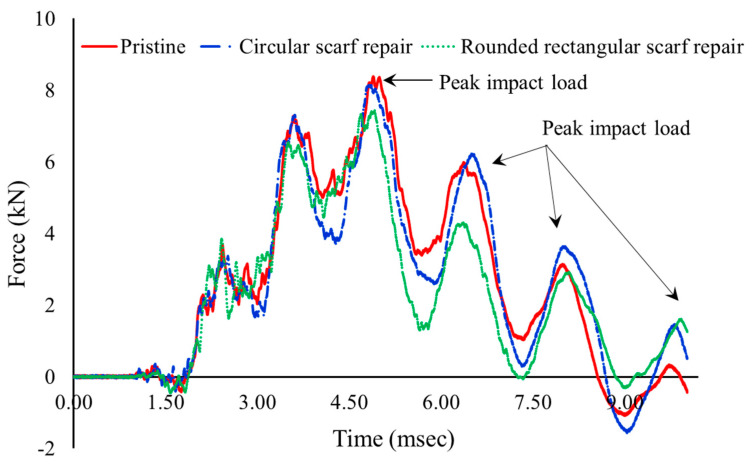
Force versus time response of impacted samples.

**Figure 4 polymers-15-02390-f004:**
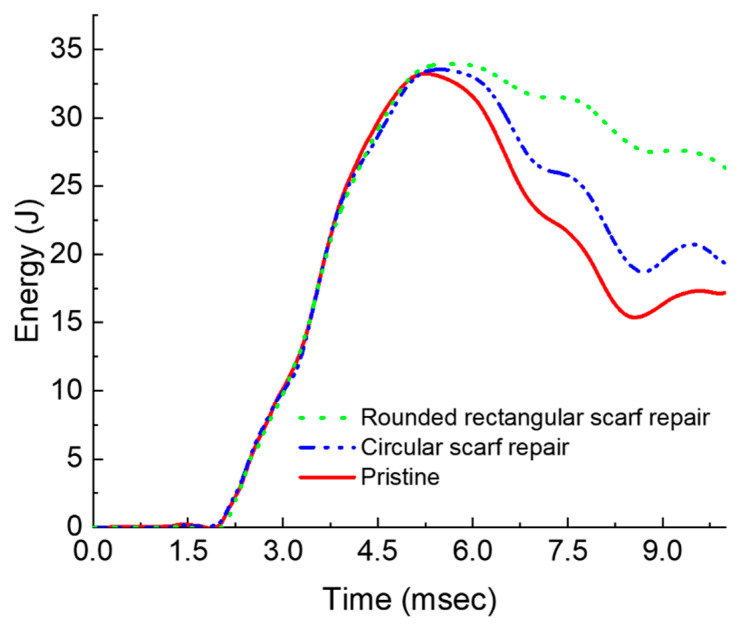
Energy versus time response of impacted laminates.

**Figure 5 polymers-15-02390-f005:**
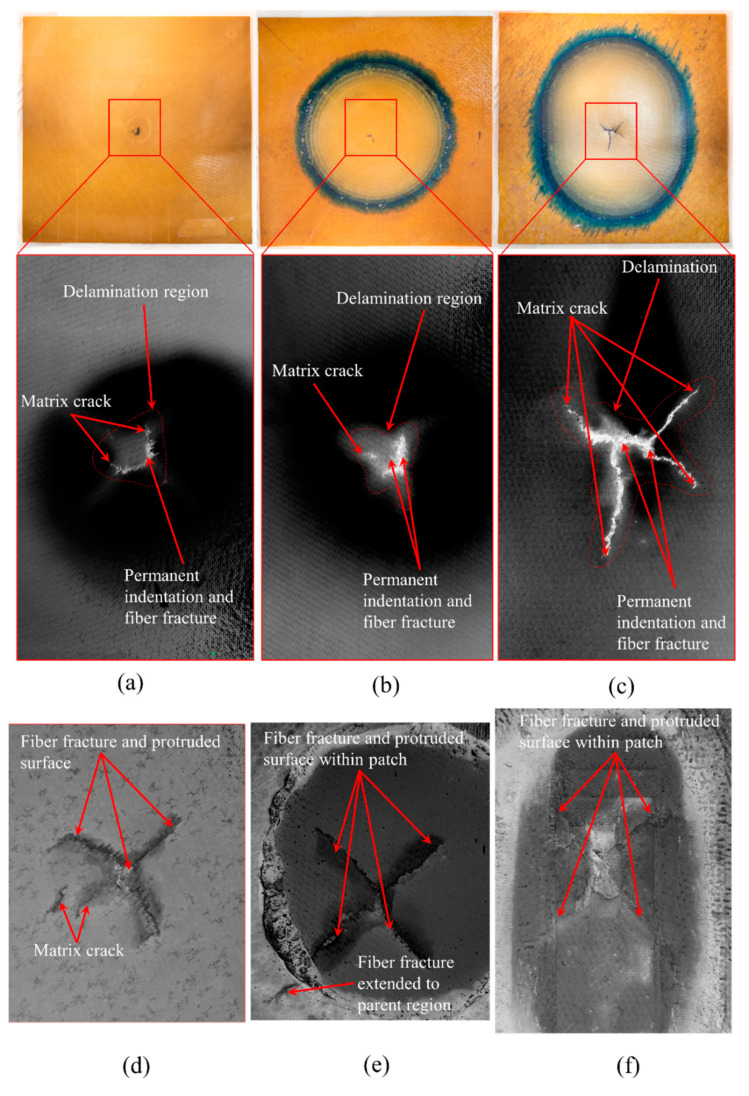
Visually observed (top view, bottom view) impact damages of (**a**,**d**) pristine laminate; (**b**,**e**) circular scarf repaired laminate; and (**c**,**f**) rounded rectangular scarf repaired laminate.

**Figure 6 polymers-15-02390-f006:**
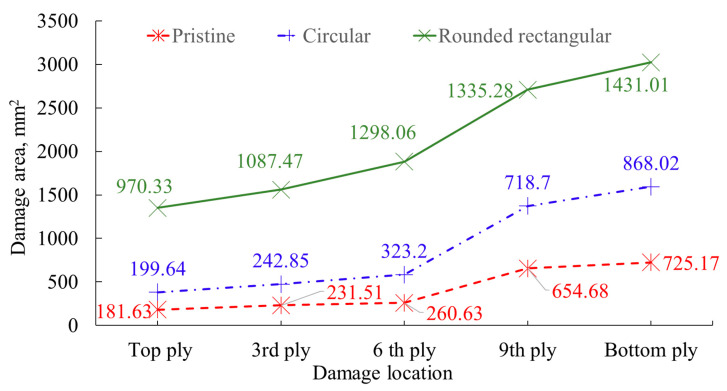
Damaged area for different plies for tested configurations.

**Figure 7 polymers-15-02390-f007:**
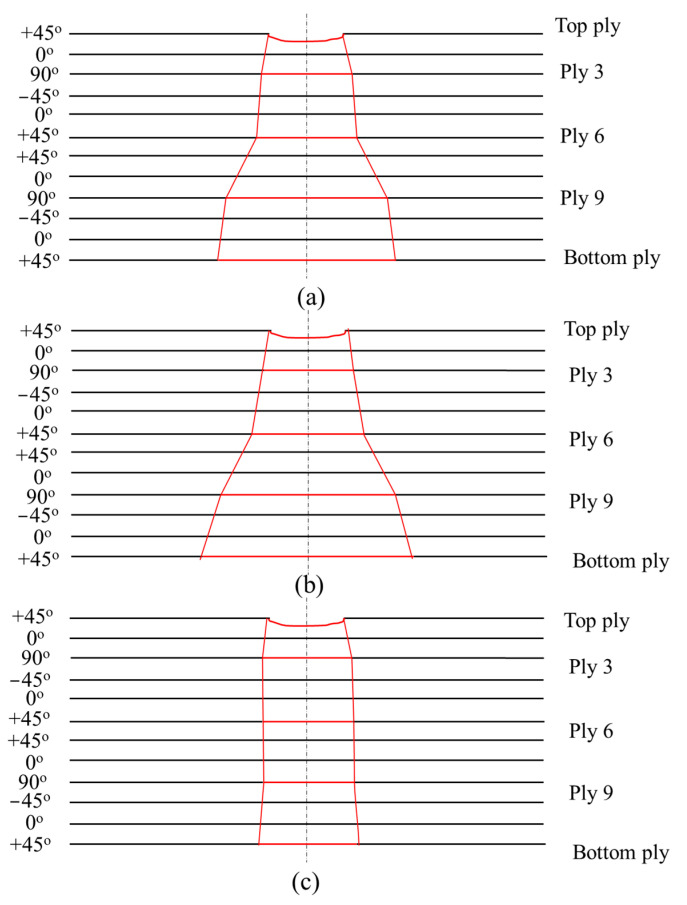
Cross-sectional damage profile with respect to the size of the top ply damage area of (**a**) pristine laminate; (**b**) circular scarf repaired laminate and (**c**) rounded rectangular scarf repaired laminate.

**Figure 8 polymers-15-02390-f008:**
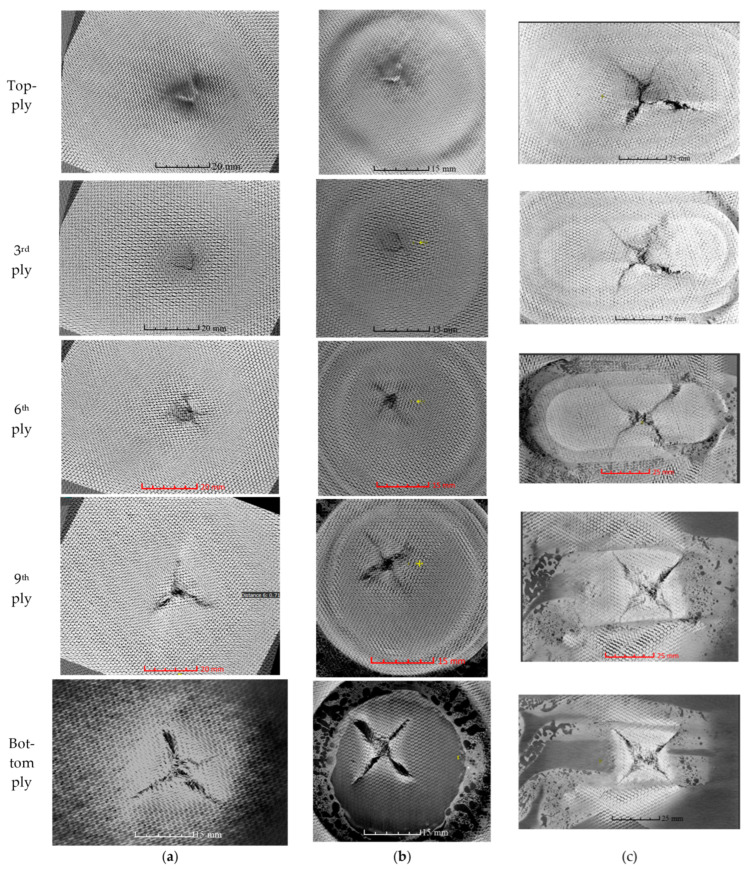
CT scan images for various plies for (**a**) pristine, (**b**) circular repair, and (**c**) rounded rectangular repair.

**Figure 9 polymers-15-02390-f009:**
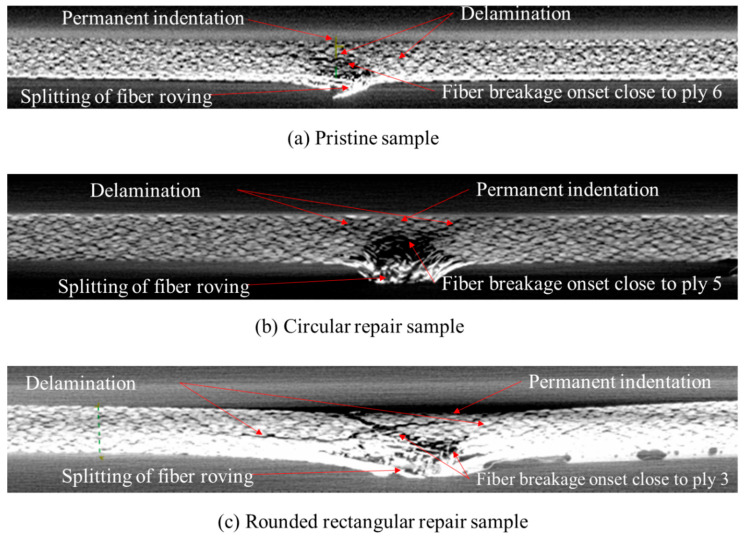
Cross-sectional view of (**a**) pristine, (**b**) circular, and (**c**) rounded rectangular repair.

**Table 1 polymers-15-02390-t001:** Peak force response of specimens.

Specimen	Pristine Sample	Circular Patch Repair			Rounded Rectangular Patch		
	A	B	C	D	E	F	G
Peak force (kN)	8.39	7.35	8.14	7.67	7.44	6.98	6.99

## Data Availability

All data generated or analyzed during this study were included in this published article and raw data will be made available upon request.

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
