# Peer review of "Effect of Scarf Repair Geometry on the Impact Performance of Aerospace Composites"

_polymers, 2023, doi:10.3390/polym15102390_

Round 1
Reviewer 1 Report
The manuscript titled “Effect of scarf repair geometry on the impact performance of aerospace composites” investigates the effect of scarf geometry in restoring the impact response. The manuscript is well written. I recommend publication of the manuscript after minor revision.
1) The central area under study is 225 x 225 mm, was any other area chosen for investigation.
2) What is the plausible reason for the rounded rectangular repaired sample to exhibit more energy absorption than the pristine and circular scarf?
3) Line 251 states that a black ink was used to fill the crack. What kind of black ink was used and why was it chosen?
4) In figure 8, the scale is not clear for 6th ply and 9th ply data.
5) Line 344, it is stated that adhesive AF3109-2K was able to resist the adhesive failure, was any flexural strength measured conducted.
Author Response
- The central area under study is 225 x 225 mm, was any other area chosen for investigation.
Response: It was chosen as part of the industrial requirement. Most of the studies experimental data is based on samples of around 150 mm x 150 mm or less with much lower impact energy.
- What is the plausible reason for the rounded rectangular repaired sample to exhibit more energy absorption than the pristine and circular scarf?
Response: Because of the asymmetricity of scarfing at the point of impact which results in a damage that is oriented largely towards the major axis of rounded rectangular scarf and shorter towards the minor axis of rounded rectangular scarf.
- Line 251 states that a black ink was used to fill the crack. What kind of black ink was used and why was it chosen?
Response: Isopropanol-based graphite black paint was used for the following reasons:
- Low viscosity of the ink able to penetrate into crevices created by crack.
- The high opacity of the ink helped in blocking high intensity light through the crack thereby helping us visualise the damage clearly.
4) In figure 8, the scale is not clear for 6th ply and 9th ply data.
Response: The scale bar in the figure is improved in the revised manuscript.
5) Line 344, it is stated that adhesive AF3109-2K was able to resist the adhesive failure, was any flexural strength measured conducted.
Response: We validated that this with:
- During CT scanning, we observed the interfacial details and found no evidence of damage propagation or relative movement between the patch and parent laminate, indicating adhesive yielding.
- With the aid of high-intensity lighting, we were able to make the same observation, aided by the distinctive blue color of the AF-3109-2k, which served as a helpful indicator.
Hope the responses are satisfactory.
Reviewer 2 Report
In this paper, the authors investigated the effect of scarf geometry in restoring the impact response of scarf patched 3 mm thick glass-fibre reinforced polymer matrix composite laminates.
The paper is properly divided in sections and sub-sections, but it needs to be revised before being considered for publication in the journal.
- The authors should check the text since some errors are present in the text;
- Which is the deviation of the data reported in table 1?
- Did the authors repeat the tests (for example 3 times) in order to assess the repeatability of the results?
Some minor errors are present in the text. Anyway, the paper is fine, clear and readable.
Author Response
- The authors should check the text since some errors are present in the text;
Response: The manuscript is thoroughly revised for technical and grammatical mistakes further.
- Which is the deviation of the data reported in table 1?
|
Specimen |
Standard deviation |
|
pristine samples |
0.13 |
|
circular scarf samples |
0.40 |
|
Rounded rectangular scarf samples |
0.26 |
- Did the authors repeat the tests (for example 3 times) in order to assess the repeatability of the results?
Response: Yes, we did repeat the tests three times and the results are consistent within the experimental scatter.
- Comments on the Quality of English Language: Some minor errors are present in the text. Anyway, the paper is fine, clear and readable.
Response: We have revised the manuscript for further technical and grammatical corrections.
Reviewer 3 Report
The manuscript presented is well written and well structured. It provides a detailed report on the effect of scarf repair geometry on the impact performance of aerospace composites. The authors have also done an excellent job of describing and explaining experimental measurements. They explained and reasoned in detail the need for this research. Overall, I recommend accepting this work in its present form.
The manuscript is well written in terms of language and style.
Author Response
We are thankful to the reviewer for the positive comments.